

# Applying auxiliary supervised depth-assisted transformer and cross modal attention fusion in monocular 3D object detection

Zhijian Wang[1], Jie Liu[2], Yixiao Sun[1], Xiang Zhou[3], Boyan Sun[1], Dehong Kong[1], Jay Xu[1], Xiaoping Yue[1] and Wenyu Zhang[1]

[1] School of Computer Science and Software Engineering, University of Science and Technology Liaoning, Anshan, Liaoning, China
[2] Anshan Power Supply Company, Liaoning Electric Power Limited Company of State Grid, Anshan, Liaoning, China
[3] Inner Mongolia Electronic Information Vocational Technical College, Huhehaote, Neimenggu, China

Corresponding author
Wenyu Zhang,
zhangwenyu8518@ustl.edu.cn

## ABSTRACT

Monocular 3D object detection is the most widely applied and challenging solution for autonomous driving, due to 2D images lacking 3D information. Existing methods are limited by inaccurate depth estimations by inequivalent supervised targets. The use of both depth and visual features also faces problems of heterogeneous fusion. In this article, we propose Depth Detection Transformer (Depth-DETR), applying auxiliary supervised depth-assisted transformer and cross modal attention fusion in monocular 3D object detection. Depth-DETR introduces two additional depth encoders besides the visual encoder. Two depth encoders are supervised by ground truth depth and bounding box respectively, working independently to complement each other's limitations and predicting more accurate target distances. Furthermore, Depth-DETR employs cross modal attention mechanisms to effectively fuse three different features. A parallel structure of two cross modal transformer is applied to fuse two depth features with visual features. Avoiding early fusion between two depth features enhances the final fused feature for better feature representations. Through multiple experimental validations, the Depth-DETR model has achieved highly competitive results in the Karlsruhe Institute of Technology and Toyota Technological Institute (KITTI) dataset, with an AP score of 17.49, representing its outstanding performance in 3D object detection.

## INTRODUCTION

As the number of vehicles increases every year, more and more traffic accidents are caused. Data from the *World Health Organization (2023)* indicates millions of people are being affected by human factor accidents like speeding and drunk driving. Autonomous driving systems are designed to avoid such accidents, enhancing road safety.

Currently, cameras and light detection and ranging (LiDAR) are the most available options for environmental perception in autonomous driving systems (*Perumal et al.,*

*2023*). However, due to the higher financial and computation cost of LiDAR, the vast majority of existing and newly manufactured vehicles are still equipped with monocular cameras for object detection (*Liu et al., 2021*). Therefore, achieving effective object detection using monocular cameras has become a significant focus and challenge for researchers.

For location-sensitive applications such as environmental perception, conventional 2D detection systems have a critical limitation, lacking the availability of providing physically correct metric information in 3D space (*Tao et al., 2023*). Recently, researchers in the field of 3D object detection have recognized the significance of depth information. Nevertheless, conventional models' network structures (*Shi et al., 2021*) (see Fig. 1A) do not place additional emphasis on such vital depth information. Given the importance of depth information, recent studies have endeavored to incorporate additional depth estimation modules into the network, aiming to extract depth information for each pixel in the image (*Rudolph et al., 2022*). However, these attempts still encounter two serious challenges.

One of the issues is the effectiveness of the depth prediction module is difficult to ensure. Typically, for deep learning networks, the training results of the network correspond to a similar supervised target. However, for additional depth modules in works like *Yan et al. (2024)*, *Chen et al. (2021)*, the depth results estimated by the module do not have a suitable supervised target to compute loss. Specifically, the final data supervision target in model training is the 3D object bounding box, which does not include depth prediction results. Therefore, the role of the current model's depth estimations method, which is supervised by the 3D bounding box, merely increases the model's depth and breadth without effectively estimating the depth within the image. Consequently, the model's ability to perceive the 3D space is not significantly improved.

Another issue is the fusion problem between depth information and visual features. Due to the integration of an additional depth prediction module on top of visual feature extraction, the model needs to fuse visual and depth features. However, visual and depth features, being heterogeneous and derived from two distinct modalities, exhibit significant differences in their data structures and feature dimensions. Simply scaling and adding these features (see Fig. 1B) would lead to mutual interference, resulting in internal information chaos within the fused features (*Xu et al., 2021*; *Kumar, Brazil & Liu, 2021*).

In order to alleviate these issues, we introduce Depth Detection Transformer (Depth-DETR), applying auxiliary supervised depth-assisted transformer and cross modal attention fusion in monocular 3D object detection. Compared to DEtection TRansformer (DETR) (*Carion et al., 2020*), Depth-DETR addresses the first identified issue by enhancing depth estimation effectiveness through the introduction of three parallel feature extraction branches in the encoder section (see Fig. 1C). These branches are tasked with extracting features from depth information supervised by ground truth depth, depth information supervised by 3D object detection boxes, and visual information, respectively. By mapping and discretizing the point cloud information corresponding to the images, ground truth depth supervision targets are obtained, thereby enabling more accurate predictions from the depth estimation model. However, some works have indicated that due to errors introduced by cross-dimensional mapping and discretization, using ground

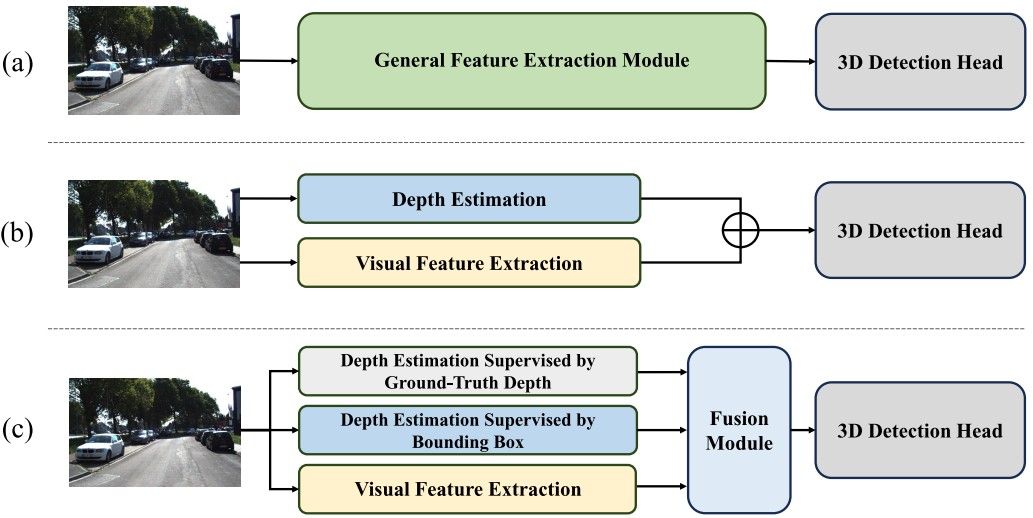

**Figure 1 Comparison of different monocular 3D object detection approaches.** (A) Indicates conventional models' network structures without emphasis on depth. (B) Indicates depth-assisted network supervised by bounding boxes. (C) Indicates Depth-DETR structure with auxiliary supervisions of ground-truth depth and specially designed fusion module.

truth depth supervision alone does not yield entirely accurate depth estimations. To compensate for these errors, Depth-DETR also introduces another depth estimation branch with a similar structure, supervised by 3D object detection boxes. The fusion of depth information obtained from these two different types of supervision provides the model with more precise depth spatial information.

To fuse three different features, this study explores various attention mechanisms for better integration of the three different features. Due to significant differences in data structure and contained information between depth features and visual features, simply scaling and adding these features would lead to mutual interference and internal chaos within the fused features. Therefore, attention mechanisms are employed to fuse these features more effectively. Addressing the shortcomings of self-attention, this study proposes a cross-modal fusion attention mechanism. By flattening features from all modalities and concatenating them together as a long 2D feature, the self-attention feature is too long to catch global attention due to we have three modalities. Cross-modal fusion attention mechanism effectively facilitates the interaction between two different modalities, generating attention weights that integrate both modalities. Ultimately, Depth-DETR introduces a parallel cross modal attention fusion mechanism. Initially, the two different depth features and the visual feature are fused into two features using cross modal attention. These two features are then combined into a single feature through dimensional stacking, followed by a final fusion using the self-attention mechanism. Consequently, the features extracted by Depth-DETR not only complement the two different depth features but also enhance the interaction and fusion between visual and depth features, enriching the final features with 3D spatial information.
With auxiliary supervision and cross modal attention fusion, Depth-DETR achieves competitive performance among other methods. On the Karlsruhe Institute of Technology and Toyota Technological Institute (KITTI) (*Geiger et al., 2013*) dataset test set, it attains $AP_{3D}$ scores of 26.38, 17.49, and 14.43 on easy, moderate, and hard difficulty levels.

## RELATED WORKS

In monocular 3D object detection, current approaches can be categorized based on how they utilize data: image-only and depth-assisted methods. Image-only detectors only work with visual features from input images, while depth-assisted methods try to better understand the 3D space with help of depth estimations from visual features. By additional depth estimation methods, depth features are generated beside visual features, leading to explorations of fusing two heterogeneous features together.

### Image-only monocular 3D object detection

Most previous monocular detectors follow the principles of conventional 2D detectors (*Ren et al., 2016*; *Detector, 2022*; *Lu et al., 2021*). Because single images inherently miss depth information, geometric consistency plays a major role in determining the location and orientation of objects. This reliance involves utilizing the known geometric properties of objects and their spatial relationships within the scene to reflect three-dimensional features from 2D information. However, the absence of direct depth cues often limits the accuracy and reliability of these predictions, making it challenging to achieve precise 3D object localization and orientation estimation.

As early works, Deep3Dbox (*Mousavian et al., 2017*) addresses orientation prediction with a novel multi-bin loss, which discretizes orientation angles for better accuracy. By using geometric priors, Deep3Dbox also enforces constraints between bounding boxes from different dimensions, ensuring consistency between the 2D projections and 3D predictions. Monocular 3D Region Proposal Network (M3D-RPN) (*Brazil & Liu, 2019*) predicts 3D object bounding boxes using 2D constraints and incorporates convolutional layers to improve the accuracy of 3D object predictions. This approach combines spatial localization guidance with depth-related feature extraction to enhance overall detection performance. Orthographic feature transform is proposed in Optical Fiber Temporal Network (OFTNet) (*Roddick, Kendall & Cipolla, 2019*). OFTNet projects image-based features into a 3D voxel space using orthographic projections, thereby improving the spatial representation of objects. By directly transforming features into a volumetric representation, OFTNet improves the model's ability to capture three-dimensional spatial information from monocular images. MonoPair (*Chen et al., 2020a*) focuses on improving detection performance by exploring spatial pairwise relationships between objects. MonoPair analyzes the orientations of targets and their positions within the scene, enhancing the context-awareness of the detection process. This spatial analysis aids in more accurately localizing and classifying objects, especially in complex visual environments. Additionally, key points are utilized in many works like *Li et al. (2020)*, *Liu, Wu & Tóth (2020)* and *Luo et al. (2021)*, as an indicator locating the targets. However, these image-only monocular methods often face challenges in accurately localizing objects

due to their exclusive dependence on 2D images, which do not provide direct depth information. This limitation can lead to challenges in precisely determining object positions and dimensions in 3D space, especially in complex scenes with varying depths and occlusions. As a comparison, our Depth-DETR adopts LiDAR input as extra supervision. With the help of accurate depth information from LiDAR features, Depth-DETR has overcome the shortage of image features only providing 2D information.

## Depth-assisted monocular 3D object detection

The inherent challenge posed by lacking of depth in images has spurred the development of depth-assisted models aimed at improving detection performance. These models leverage additional information from depth estimation networks or employ geometric reasoning techniques to enhance the accuracy of object localization and classification.

Depth-guided Dynamic-Depthwise-Dilated Local Convolutional Network (D4LCN) (*Ding et al., 2020*) and Depth-Conditioned Dynamic Message Propagation for Monocular 3D (DDMP-3D) (*Yin, Zhou & Krahenbuhl, 2021*) pay attention to approaches based on fusion. These approaches integrate images with estimated depth information using convolutional networks. These methods aim to leverage the complementary strengths of 2D image features and depth cues to enhance 3D object detection performance. By combining these data sources, D4LCN and DDMP-3D upgrade the accuracy and robustness of object localization and classification in three-dimensional space. Although both models incorporate additional depth estimation modules to assist in acquiring more geometric information, the ultimate supervised target remains the detection box. The predicted results from the depth estimation module cannot be effectively validated. Inaccurate depth estimation results can indeed impact the overall effectiveness of the model. To address this issue approaches such as Categorical Depth Distribution Network (CaDDN) (*Reading et al., 2021*) and Monocular 3D Object Detection with Depth-Aware Transformer (MonoDTR) (*Huang et al., 2022*) employ additional supervised targets. These methods integrate supplementary supervised learning objectives to increase the accuracy and reliability of monocular object detection. CaDDN focuses on learning categorical depth distributions to generate bird's-eye-view features. By leveraging these features, CaDDN recovers bounding boxes from the projected view, improving the accuracy of object localization and orientation estimation in monocular settings. MonoDTR generates ground truth depth supervision by leveraging LiDAR data. This approach enhances monocular 3D object detection by utilizing precise depth information obtained from point cloud data, which is then used to supervise and increase the accuracy of depth estimation in the detection process. Recently, Depth Equivariant Network for Monocular 3D Object Detection (DEVIANT) (*Kumar et al., 2022*), A Unified Vehicle and Infrastructure-side Monocular 3D Object (MonoUNI) (*Jinrang, Li & Shi, 2024*) and some other works (*Qin & Li, 2022*) explored the approach of depth translations in the projective manifold, showing greater abilities of cross-datasets. SeaBird (*Kumar et al., 2024*) emphasized the ability of detecting large objects, effectively integrating BEV segmentation on foreground objects for 3D detection, with the segmentation head trained with the dice loss. However, projecting LiDAR and other 3D data to pixel-level depth supervisions is relying on pre-trained

projecting models. These pre-trained models cannot guarantee absolute accuracy, and incorrect supervision targets will similarly lead to deviations in depth estimation from real results, thereby affecting the final accuracy of the model. Depth-DETR utilized depth information in the other way. Extra supervisions of depth estimator could provide more accurate depth feature, then enrich the final features. While other methods of utilize depth estimator could provide inaccurate depth informations.

## Fusion methods in multi-modal 3D object detection

With the development of different sensors, 3D object detection is moving forward to multimodality. Comparing to single-modal models, multimodal models are able to utilize images, depth information, point clouds, and other type of data together for object detection. However, this approach faces challenges of integrating multimodal data. Instead of tacking or adding features together, an increasing number of researches is focusing on how to effectively and efficiently fuse multimodal data using attentions.

Single-stream multimodal Transformer like Universal Image-text Representation Learning (UNITER) (*Chen et al., 2020b*) and Vision-and-Language Transformer (ViLT) (*Kim, Son & Kim, 2021*) typically consists of a series of multimodal self-attention modules at each layer. These models facilitate early and extensive interaction and fusion of different modalities. Tokens representing different modalities are combined through concatenation. Each layer's attention mechanisms can focus on both intra-modal (within-modal) and inter-modal (across-modal) contexts. This approach enables the model to effectively utilize and integrate information from various modalities throughout the network's depth, enhancing its ability to capture comprehensive contextual dependencies both within and across modalities. However, flattening and concatenating multimodal features can result in excessively long feature vectors, which negatively impacts computational efficiency. In order to overcome the disadvantages of self-attention, MonoDETR (*Zhang et al., 2023*) and Bird's-eye View representations with spatiotemporal transformers (BEVFormer) (*Li et al., 2022*) employ cross-attention mechanisms to combine multi-modalities features. In the cross-attention mechanism, queries from one modality are used to attend to keys and values from another modality, facilitating the exchange of information between the two modalities. Following this initial fusion, self-attention layers are applied to the merged features for further refinement, allowing the model to deeply integrate and contextualize the combined information. Despite the advantages of cross-attention in merging heterogeneous features, this method has some limitations. The fusion directions in cross-attention are predetermined and fixed, which can restrict the interaction strength between distant modalities. This rigidity can limit the model's ability to dynamically adapt to varying degrees of relevance and dependency between the features of different modalities. Depth-DETR introduces a parallel cross modal attention fusion mechanism. With this design, the features extracted by Depth-DETR not only complement the two different depth features but also enhance the interaction and fusion between visual and depth features, enriching the final features with 3D spatial information.

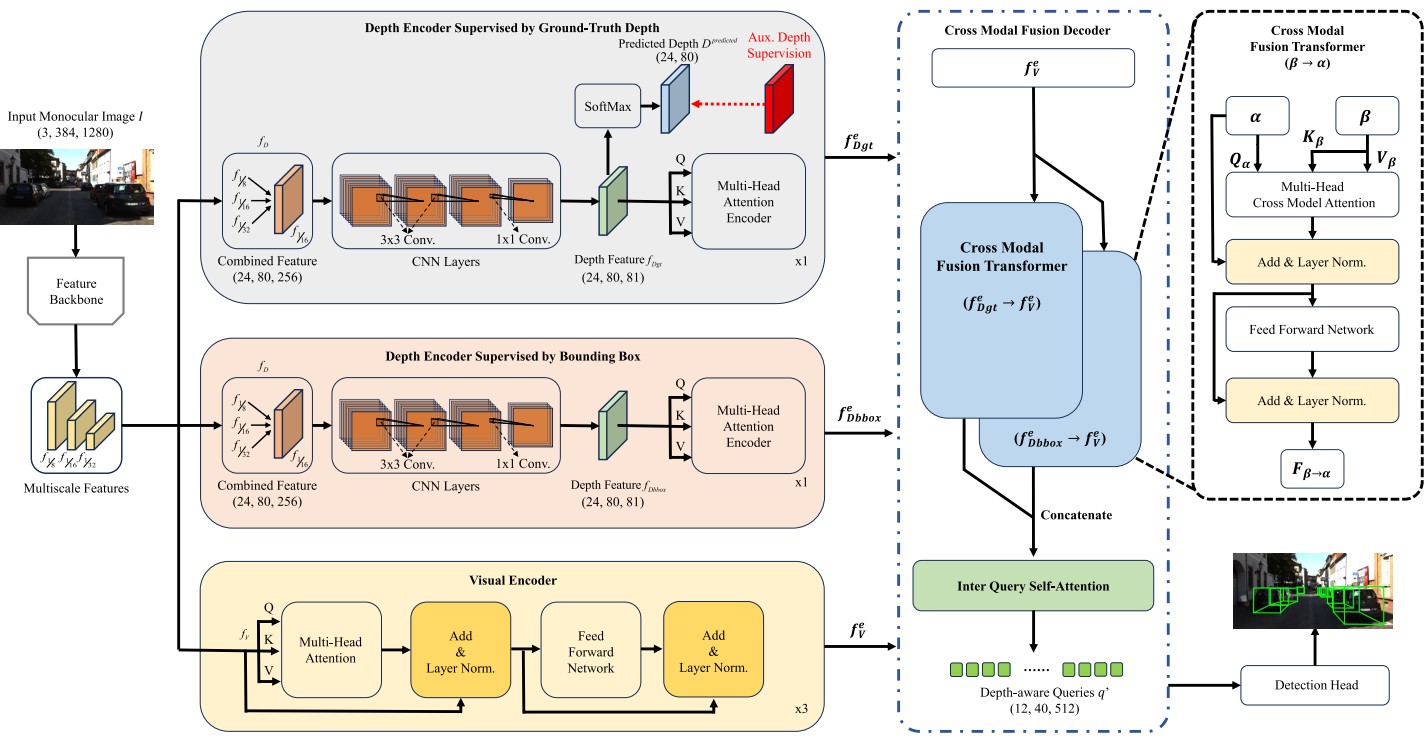

**Figure 2 Overall structure of Depth-DETR.** The monocular image firstly goes through extration backbones to obtain multiscale features. Then, these feautures are processed by three approaches for depth features supervised by ground true depth, depth features supervised by 3D object detection boxes, and visual features. Two parallel cross modal fusion transformer fuses two depth features with visual features, following by one self-attention module outputting the final feature. Finally, a 3D detection head generates bounding boxes upon the final feaure.

## METHODS

The overall framework of Depth-DETR is shown in Fig. 2. The concurrent visual and depth feature extraction is introduced, with the detail work of generating ground true depth supervisions from point cloud data. Then, we introduce the depth-assisted transformer encoder, which contains three parallel approaches for depth features supervised by ground true depth, depth features supervised by 3D object detection boxes, and visual features. Thirdly, we illustrate cross modal fusion transformer as the decoder of Depth-DETR, fusing outputs of three parallel encoders. Finally, we introduce the attribute prediction and loss functions.

### Feature extraction

The image $I \in \mathbb{R}^{H \times W \times 3}$ is inputted into the network with its height and width. Depth-DETR adoptes feature backbone to generate four levels of feature maps in size $\frac{1}{4}, \frac{1}{8}, \frac{1}{16}$, and $\frac{1}{32}$ of original input image size.

#### Visual and depth features

The features $f_V$ with size of $12 \times 40 \times C$, is the input of visual encoder. In this case, we regard the $f_{\frac{1}{32}}$ as $f_V$. As for the depth features, three-level features $f_{\frac{1}{8}}, f_{\frac{1}{16}}$ and $f_{\frac{1}{32}}$ from

backbone is selected to generate the depth features $f_D$. By down sampling $f_{\frac{1}{8}}$ via $3 \times 3$ convolutional layers and up sampling $f_{\frac{1}{32}}$ via bilinear interpolation, two features are both reshaped to $24 \times 80 \times C$. Then two reshaped features and $f_{\frac{1}{16}}$ are fused by element-wise addition as the depth features $f_D$.

### Auxiliary depth supervision

In one of the depth encoders, the depth prediction $D^{pred}$ is supervised by ground-truth $D^{gt}$ calcualted from LiDAR data $P$. To obtain the $D^{gt}$ from the LiDAR coordinate to the camera coordinate, we performers:

$$P^{img}(u, v, d) = K(RP + t), \tag{1}$$

where $r$ and $t$ as the rotation and translation matrix, and $K$ as the camera intrinsic parameter. By utilizing minimum pooling and one hot coding on $P^{img}$, we now have the $D^{gt}$ as the ground-truth depth supervision.

With continuous $D^{gt}$ data, we then use linear-increasing discretization to discretize $D^{gt}$ to separated depth bins (*Tang, Dorn & Savani, 2020*).

$$d = d_{\min} + \frac{d_{\max} - d_{\min}}{D(D+1)} \cdot i(i+1), \tag{2}$$

where $i$ is the bin index from 1 to $D$, $D$ represents the number of depth bins $d_{\min}$, and $d_{\max}$ represents the range of depth.

## Depth-assisted transformer encoder

The depth-assisted encoder of Depth-DETR is composed of a depth encoder supervised by ground-truth depth, a depth encoder supervised by bounding boxes, and a visual encoder.

### Depth encoder supervised by ground-truth depth

Given depth feature $f_D$, two $3 \times 3$ convolutional layers with padding of 2 is applied to extract the geometrical information. After adapting the channels size in a $1 \times 1$ convolutional layer, we have the extracted depth feature, denoted as $f_{Dgt}$ in shape of $24 \times 80 \times 81$. Then, $f_{Dgt}$ goes through a transformer encoder block. Self-attention layers and feed-forward networks are utilized, finding dependencies of depth information from different areas.

To supervised by ground-truth depth, $f_{DG}$ is softmaxed to a predicted depth $D^{predicted}$, which will be supervised by $D^{gt}$.

### Depth encoder supervised by bounding boxes

The $D^{gt}$ generated by point cloud data is limited by the 3D to 2D projections. As a result, depth feature $f^e_{Dgt}$ suffers from errors by dimension projections, causing effectiveness loss in object detection. To overcome is issue, we reserve the depth encoder supervised by bounding boxes to extract the depth feature $f^e_{Dbbox}$, with similar convolutional and self-attention structures. Even if two different depth encoder have their own limitations, fusing together can make up for the shortcomings of both parties involved.

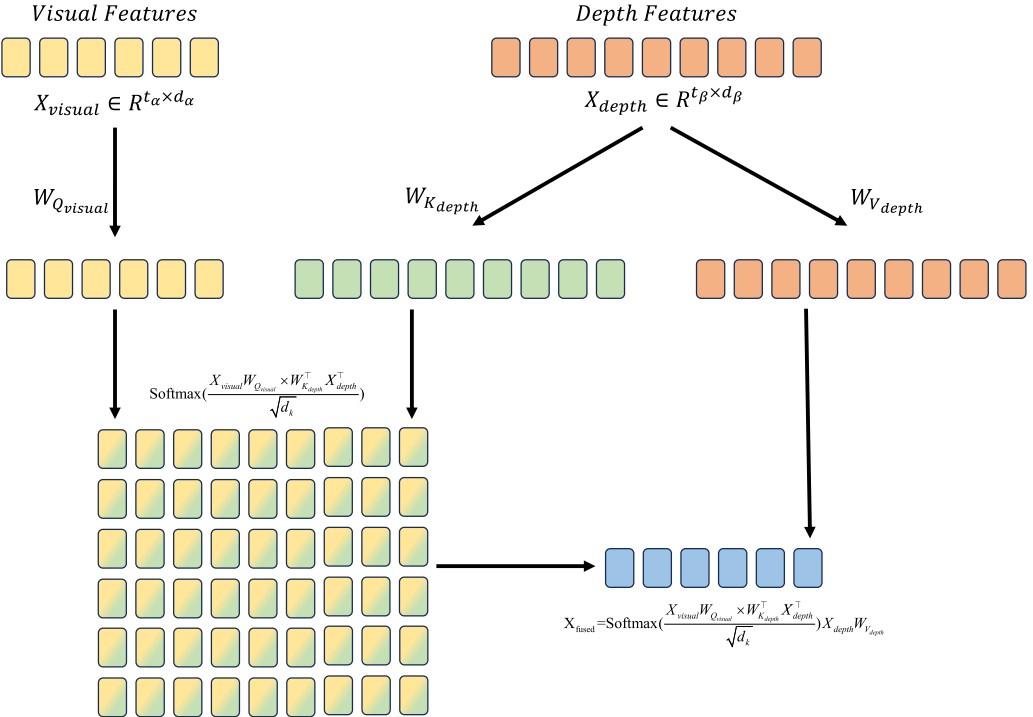

**Figure 3 Illustration for cross modal attention mechanism.** Features from different modalities are transformed into $Q$, $K$ and $V$, the attention score is calcualted by $Q$ from visual features and $K$ from depth features, then applying softmax to generate attention weights. The fused feature is generated by attention weights and $V$ from depth features.

### Visual encoder

Given depth feature $f_V$, a transformer encoder block is adopted, similar to the one utilized in the depth encoder. Instead of one block, the visual encoder adopts three encoder block to generate scene-level embeddings with global receptive fields, denoted as $f_V^e$.

## Decoder of Depth-DETR

As three features input to the decoder of Depth-DETR, we first combine depth feature $f_{Dgt}^e$ and $f_{Dbbox}^e$ with $f_V^e$ respectively. In order to fuse depth features and visual features, which are different modalities with different feature sizes. We first introduce cross modal attention.

### Cross modal attention

To combine two modalities which are visual and depth, the features are defined as $X_{visual} \in \mathbb{R}^{t_\alpha \times d_\alpha}$ and $X_{depth} \in \mathbb{R}^{t_\beta \times d_\beta}$, where $t_\alpha = H_\alpha \times W_\alpha$ and $t_\beta = H_\beta \times W_\beta$.

$W_{Q_{visual}}$, $W_{K_{depth}}$ and $W_{V_{depth}}$ are defined as the learnable transformation matrix in attention mechanism. Therefore, the fusion process from depth to visual feature is shown as:

$$X_{fused} = \text{softmax}\left(\frac{X_{visual}W_{Q_{visual}} \times W_{K_{depth}}^\top X_{depth}^\top}{\sqrt{d_k}}\right)X_{depth}W_{V_{depth}}. \tag{3}$$

We call (Eq. (3)) a single-head cross modal attention, which is illustrated in Fig. 3.

### Cross modal fusion

On the top of cross modal attention blocks, cross modal fusion is proposed in this section. Cross modal fusion mergers two modalities together, by projecting information from one modality to the other. Initially, two depth features $f_{Dgt}^e$ and $f_{Dbbox}^e$ are fused with the visual features $f_V^e$. To ensure that each input element is sufficiently aware of its neighboring elements, we pass the input features through a $1 \times 1$ convolutional layer. This step guarantees that both depth and visual features have the same channel size.

For example of fusing $f_{Dgt}^e$ to $f_V^e$, denoted by $f_{Dgt}^e \to f_V^e$. Cross modal fusion contains $N_{cross}$ layers of cross modal attention blocks, illustrated in right hand side of Fig. 2, where $f_{Dgt}^e$ and $f_V^e$ are fused by a multi-head version of Eq. (4). After layer normalizations and feed-forward network, $f_{Dgt \to V}^e$ is generated. Same process for $f_{Dbbox}^e \to f_V^e$, generating $f_{Dbbox \to V}^e$.

Two fused features $f_{Dgt \to V}^e$ and $f_{Dbbox \to V}^e$ then concatenate together by channel dimension. With the help of self-attention layers, all three feature finally combined to depth-aware queries $q'$.

### Depth positional encodings

In order to maximize the performance of depth information, a learnable depth positional encoding is introduced in the cross modal attention for $f_{Dgt}^e$ and $f_{Dbbox}^e$, moving away from traditional sinusoidal functions. We achieve this by creating positional encoding $p_D \in \mathbb{R}^{(d_{max}-d_{min}+1) \times C}$, with each row encoding depth positional information for each meter between $d_{min}$ and $d_{max}$.

For each pixel $(x, y)$ in $D^{predicted}$, we derive its $(k + 1)$-categorical depth prediction confidence. The depth estimate for pixel $(x, y)$ is then obtained through a weighted summation of these depth-bin confidences and their corresponding depth values, as expressed in the following equation:

$$d_{map}(x, y) = \sum_{i=1}^{k+1} D^{predicted}(x, y)[i] \cdot d_{bin}^i, \tag{4}$$

where $d_{bin}^i$ denotes the starting value of the $i$-th depth bin. Then, we linearly interpolate positional encodings $p_D$ according to the depth $d_{map}(x, y)$ to obtain the depth positional encoding for the pixel $(x, y)$. By pixel-wisely adding $p_D$ with $f_{Dgt}^e$ and $f_{Dbbox}^e$, fused queries are able to understand 3D geometry better in the cross modal attention layer.

## Detection heads and loss

After the decoder, the queries are processed through heads combined by MLP layers to predict various 3D attributes, including object category, 2D size, projected 3D center, depth, 3D size, and orientation.

The losses of these six attributes are categorized by two groups. As these attributes primarily pertain to the 2D visual appearance of the image, object category, 2D size, and

the projected 3D center are included in the first group $\mathscr{L}2D$. As a contrary, 3D spatial properties of the object like depth, 3D size, and orientation are set as the other group $\mathscr{L}3D$.

In detection transformer approaches, the number of queries is set as $N$, and $N_{gt}$ represents the number of ground-truth objects. We firstly match $N_{gt}$ valid pairs out of $N$, and the overall loss is formulated as:

$$\mathscr{L}_{overall} = \frac{1}{N_{gt}} \cdot \sum_{n=1}^{N_{gt}} (\mathscr{L}_{2D} + \mathscr{L}_{3D}) + \mathscr{L}_{dmap}, \tag{5}$$

where $\mathscr{L}_{dmap}$ represents the Focal loss (*Lin et al., 2017*) of the predicted depth map $D^{predicted}$.

## EXPERIMENTS

### Settings

#### Dataset and implementation details

KITTI dataset (*Geiger et al., 2013*) is one of most well-known benchmark in monocular 3D object detection works, having 7,481 and 7,518 images in its training and testing sets. Following the work in *Chen et al. (2016)*, we create a validation set of 3,769 images from the training data. The performance of our network is evaluated in easy, moderate, and hard difficulty. This difficulty is categorized based on the distance and occlusion of the target object by KITTI officially. Average precision ($AP$) at 40 recall positions is adopted in these experiments as evaluation metrics, for both 3D bounding boxes $AP_{3D}$ and bird's-eye view bounding boxes $AP_{BEV}$.

nuScenes (*Caesar et al., 2020*) comprises 28,130 training and 6,019 validation images captured from the front camera. We use validation split for cross-dataset evaluation.

Training Depth-DETR on a single A100 GPU takes 195 epochs, with a batch size of 16 and a learning rate of $2 \times 10^{-4}$. The AdamW optimizer with a weight decay of $10^{-4}$, and reduce the learning rate by a factor of 0.1 at epochs 125 and 165 are used as training strategies.

ResNet-50 (*He et al., 2016*) is adopted as the backbone in feature extraction stages. We set the multi-head attention to eight heads and the number of queries $N$ to 50. For depth supervision, $D$ is set to be 70, $d_{\min}$, and $d_{\max}$ are set to be 1 to 81.

### Comparison

#### Performance

In Table 1, Depth-DETR demonstrates outstanding performance on both the KITTI test and validation sets. On the test set, our approach surpasses other methods, particularly at the moderate difficulty level, which is the most critical metric in the benchmark. Depth-DETR outperforms all existing methods, achieving higher scores than the second-best methods by +1.38%, +1.02%, and +0.85% in $AP_{3D}$, and by +1.54%, +1.17%, and +0.99% in $AP_{BEV}$ across the three difficulty levels.

Moreover, Depth-DETR significantly outperforms other depth-assisted methods. For example, compared to the top depth-assisted methods like MonoDTR, Depth-DETR achieves improvements of +4.39%, +7.90%, and +1.70% in $AP_{3D}$. This superior

**Table 1 Comparison of the latest method performance on KITTI dataset in car category.**

| Method | Test, $AP_{3D}$ | | | Test, $AP_{BEV}$ | | | Val, $AP_{3D}$ | | |
|---|---|---|---|---|---|---|---|---|---|
| | Easy | Mod. | Hard | Easy | Mod. | Hard | Easy | Mod. | Hard |
| D4LCN | 16.65 | 11.72 | 9.51 | 22.51 | 16.02 | 12.55 | – | – | – |
| DDMP-3D | 19.71 | 12.78 | 9.80 | 28.08 | 17.89 | 13.44 | – | – | – |
| CaDDN | 19.17 | 13.41 | 11.46 | 27.94 | 18.91 | 17.19 | 23.57 | 16.31 | 13.84 |
| MonoDTR | 21.99 | 15.39 | 12.73 | 28.59 | 20.38 | 17.14 | 24.52 | 18.57 | 15.51 |
| AutoShape | 22.47 | 14.17 | 11.36 | 30.66 | 20.08 | 15.59 | 20.09 | 14.65 | 12.07 |
| SMOKE | 14.03 | 9.76 | 7.84 | 20.83 | 14.49 | 12.75 | 14.76 | 12.85 | 11.50 |
| PGD | 19.05 | 11.76 | 9.39 | 26.89 | 16.51 | 13.49 | 19.27 | 13.23 | 10.65 |
| MonoDLE | 17.23 | 12.26 | 10.29 | 24.79 | 18.89 | 16.00 | 17.45 | 13.66 | 11.68 |
| MonoRCNN | 18.36 | 12.65 | 10.03 | 25.48 | 18.11 | 14.10 | 16.61 | 13.19 | 10.65 |
| GrooMeD-NMS | 18.10 | 12.32 | 9.65 | 26.19 | 18.27 | 14.05 | 19.67 | 14.32 | 11.27 |
| MonoFlex | 19.94 | 13.89 | 12.07 | 28.23 | 19.75 | 16.89 | 23.64 | 17.51 | 14.83 |
| GUPNet | 20.11 | 14.20 | 11.77 | – | – | – | 22.76 | 16.46 | 13.72 |
| MonoGround | 21.37 | 14.36 | 12.62 | 30.07 | 20.47 | 17.74 | 19.67 | 14.32 | 11.27 |
| DEVIANT | 21.88 | 14.46 | 11.89 | 29.65 | 20.44 | 17.43 | 25.24 | 18.69 | 15.58 |
| MonoDETR | 25.00 | 16.47 | 13.58 | 33.60 | 22.11 | 18.60 | 28.84 | 20.61 | 16.38 |
| Depth-DETR (Ours) | 26.38 | 17.49 | 14.43 | 35.14 | 23.28 | 19.59 | 30.32 | 21.88 | 17.57 |
| *Improvement* | +1.38 | +1.02 | +0.85 | +1.54 | +1.17 | +0.99 | +1.46 | +1.27 | +1.19 |

**Table 2 Cross-dataset evaluation of the KITTI *val* model on and nuScenes frontal *val* cars with depth MAE.**

| Method | KITTI *val* | | | | nuScenes frontal *val* | | | |
|---|---|---|---|---|---|---|---|---|
| | $0-20$ | $20-40$ | $40-\infty$ | All | $0-20$ | $20-40$ | $40-\infty$ | All |
| M3D-RPN | 0.56 | 1.33 | 2.73 | 1.26 | 0.94 | 3.06 | 10.36 | 2.67 |
| MonoRCNN | 0.46 | 1.27 | 2.59 | 1.14 | 0.94 | 2.84 | 8.65 | 2.39 |
| GUP Net | 0.45 | 1.10 | 1.85 | 0.89 | 0.82 | 1.70 | 6.20 | 1.45 |
| DEVIANT | 0.40 | 1.09 | 1.80 | 0.87 | 0.76 | 1.60 | 4.50 | 1.26 |
| MonoUNI | 0.38 | 0.92 | 1.79 | 0.865 | 0.72 | 1.79 | 4.98 | 1.43 |
| Depth-DETR (Ours) | 0.372 | 0.90 | 1.77 | 0.85 | 0.74 | 1.57 | 4.55 | 1.31 |

performance is achieved without relying on handcrafted designs, highlighting the simplicity and effectiveness of our approach.

To show the cross-dataset performance of Depth-DETR, Table 2 lists the result of our KITTI val model on the KITTI val and nuScenes frontal val images, using mean absolute error (MAE) of the depth (*Shi et al., 2021*). Depth-DETR is better than Geometry Uncertainty Projection Network (GUPNet) and achieves similar competitive performance to DEVIANT and MonoUNI. While DEVIANT and MonoUNI are equivariant to the depth translations and are more robust to data distribution changes, Depth-DETR is limited by the ground-truth depth supervision provided by KITTI.

**Table 3 Performance of the car category on the KITTI validation set.**

| Method | $AP_{3D}$ @IOU = 0.7 | | | $AP_{BEV}$ @IOU = 0.7 | | | $AP_{3D}$ @IOU = 0.5 | | | $AP_{BEV}$ @IOU = 0.5 | | |
|---|---|---|---|---|---|---|---|---|---|---|---|---|
| | Easy | Mod. | Hard | Easy | Mod. | Hard | Easy | Mod. | Hard | Easy | Mod. | Hard |
| CenterNet | 0.60 | 0.66 | 0.77 | 3.46 | 3.31 | 3.21 | 20.00 | 17.50 | 15.57 | 34.36 | 27.91 | 24.65 |
| MonoGRNet | 11.90 | 7.56 | 5.76 | 19.72 | 12.81 | 10.15 | 47.59 | 32.28 | 25.50 | 48.53 | 35.94 | 28.59 |
| MonoDIS | 11.06 | 7.60 | 6.37 | 18.45 | 12.58 | 10.66 | – | – | | – | – | |
| M3D-RPN | 14.53 | 11.07 | 8.65 | 20.85 | 15.62 | 11.88 | 48.53 | 35.94 | 28.59 | 53.35 | 39.60 | 31.76 |
| MonoPair | 16.28 | 12.30 | 10.42 | 24.12 | 18.17 | 15.76 | 55.38 | 42.39 | 37.99 | 61.06 | 47.63 | 41.92 |
| MonoDLE | 17.45 | 13.66 | 11.68 | 24.97 | 19.33 | 17.01 | 55.41 | 43.42 | 37.81 | 60.73 | 46.87 | 41.89 |
| GUP Net | 22.76 | 16.46 | 13.72 | 31.07 | 22.94 | 19.75 | 57.62 | 42.33 | 37.59 | 61.78 | 47.06 | 40.88 |
| Depth-DETR (Ours) | 30.32 | 21.88 | 17.57 | 40.01 | 29.72 | 23.41 | 67.36 | 53.71 | 49.09 | 69.82 | 56.14 | 51.55 |

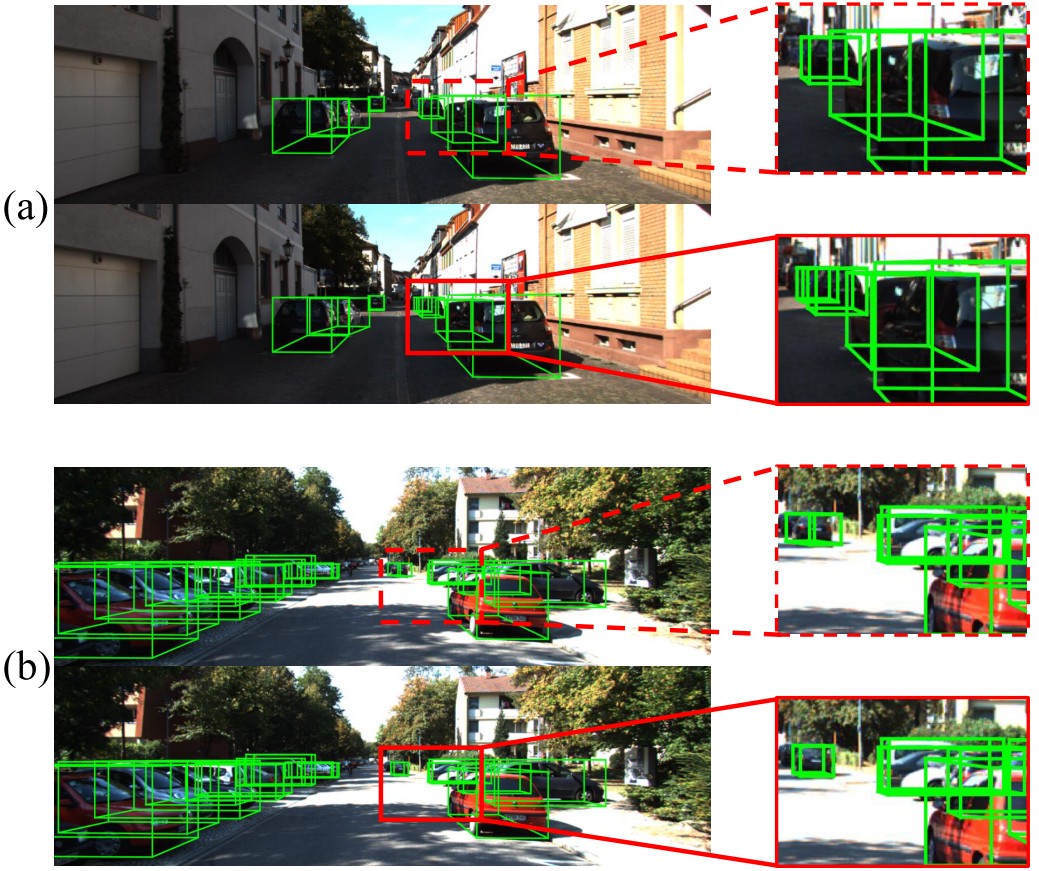

**Figure 4 Comparsion in visualizations.** Green bounding boxes indicate the detection result of baseline model (A) and Depth-DETR (B). Red boxes are marked as differences between two models.

We also present our model's performance on the KITTI validation set in Table 3 for better comparison, including different tasks and Intersection over Union (IoU) thresholds. Specifically, our method gets the best performance as competing method GUP Net at the both 0.5 and 0.7 IoU threshold.

**Table 4 Comparsions between feature inputs on KITTI *val* sets.**

| $f_V^e$ | $f_{Dgt}^e$ | $f_{Dbbox}^e$ | Val, $AP_{3D}$ | | |
|---|---|---|---|---|---|
| | | | **Easy** | **Mod.** | **Hard** |
| ✓ | | | 18.95 | 14.48 | 12.87 |
| | ✓ | | 16.28 | 12.30 | 11.51 |
| | | ✓ | 14.76 | 12.85 | 11.05 |
| ✓ | ✓ | | 28.84 | 20.61 | 16.38 |
| ✓ | | ✓ | 22.76 | 16.46 | 13.72 |
| ✓ | ✓ | ✓ | 30.32 | 21.88 | 17.57 |

**Table 5 Different fusion methods on KITTI *val* sets.**

| Fusion method | Val, $AP_{3D}$ | | |
|---|---|---|---|
| | **Easy** | **Mod.** | **Hard** |
| Concatenate self-attention fusion | 28.64 | 19.82 | 15.46 |
| Series of cross attention fusion | 29.02 | 20.08 | 16.68 |
| Depth-DETR (Cross modal attention) | 30.32 | 21.88 | 17.57 |

*Visualization*

As depicted in Fig. 4, examples from the KITTI test dataset illustrate our findings. In the figure, the green bounding boxes denote the detected objects, while the red dashed rectangles highlight false and missed detections, and the red solid rectangles indicate correct detections. Compared to our baseline on the top, Depth-DETR leverages depth encoders and cross-modal attention to enhance key feature information in local details of challenging objects during detection as a comparison in the bottom. This underscores the effectiveness of incorporating supervised depth information and a refined fusion mechanism, which improves the model's capability to understand three-dimensional spatial information within images.

## Ablation study

As the core depth encoder, we explore how to better depth encoder to interact with final feature in Table 4, by only utilized one or two of the encoders. Because of the deeper transformer network, the visual feature is the feature contains more information among three features, while depth encoders are limited by the lightweight design of network. Therefore, if the network only takes one feature to perform 3D object detection, the visual feature will receive the best performance. The $AP_{3D}$ of visual feature is 18.95, improved by +2.67% and +4.19% comparing to other two depth features.

Although both depth encoder work well with the visual features, depth feature supervised by ground-truth depth performs better due to its extra supervision. As shown in the result, two types of depth feature have different draws of their own. One due to the inaccurate point cloud projections, the other due to the unequal supervise targets.

**Table 6 The design of depth positional encodings.**

| Positional encoding | Val, $AP_{3D}$ | | |
|---|---|---|---|
| | Easy | Mod. | Hard |
| w/o | 26.76 | 18.94 | 15.85 |
| 2D sin/cos | 26.48 | 18.63 | 15.52 |
| Depth-DETR (learnable depth pe) | 30.32 | 21.88 | 17.57 |

**Table 7 Five different runs on KITTI Val cars.**

| Method | Run | $AP_{3D}$ @IOU = 0.7 | | | $AP_{BEV}$ @IOU = 0.7 | | |
|---|---|---|---|---|---|---|---|
| | | Easy | Mod. | Hard | Easy | Mod. | Hard |
| MonoDETR | 1 | 28.63 | 19.93 | 17.34 | 37.51 | 27.11 | 23.33 |
| | 2 | 28.15 | 20.76 | 16.57 | 37.30 | 27.01 | 23.24 |
| | 3 | 28.28 | 19.81 | 17.25 | 37.15 | 26.90 | 23.16 |
| | 4 | 28.78 | 20.83 | 17.47 | 37.13 | 26.94 | 23.20 |
| | 5 | 28.84 | 20.61 | 16.38 | 37.57 | 27.24 | 23.35 |
| | Avg | 28.53 | 20.38 | 17.00 | 37.33 | 27.04 | 23.25 |
| **Depth-DETR (Ours)** | 1 | 30.10 | 21.16 | 16.98 | 39.98 | 29.54 | 23.33 |
| | 2 | 30.24 | 21.87 | 17.86 | 40.26 | 29.88 | 23.54 |
| | 3 | 29.96 | 20.49 | 16.94 | 39.03 | 28.99 | 22.67 |
| | 4 | 30.45 | 20.67 | 17.32 | 39.92 | 29.25 | 22.95 |
| | 5 | 30.32 | 21.88 | 17.57 | 40.01 | 29.72 | 23.41 |
| | Avg | 30.21 | 21.21 | 17.33 | 39.84 | 29.47 | 23.18 |

However, combining two depth features together can balances each others limitations and generate more effective performance. In the result, combining three features together could give an increase up to +7.56%.

In Table 5, we experiment three different fusion methods. Self-attention, as a method for extracting information from a single feature, requires flattening and concatenating the three different modality features into a new feature. When visual features are the primary focus and the two depth features are secondary, the concatenated feature becomes excessively long and contains too much irrelevant information. Because of this drawback, using self-attention to fuse features only results in 28.64 in $AP_{3D}$. Stacking cross-attention and self-attention together could solve this problem, but manually setting the sequence of fusion does make it hard to find attention connections among modalities, especially modalities which do not perform cross-attention directly. Therefore, stacked cross attention was receives improvement of +0.38%. However, since cross-modal fusion attention can only fuse data from two different modalities at a time, sequential fusion is necessary when dealing with three modalities. The fixed order of sequential fusion results in lower interaction intensity between distant modalities, hindering effective integration of data from all three modalities. As a comparison, using cross modal attention to parallel

fuse two depth features with visual features helps to avoid early interactions. This method with 30.32 in $AP_{3D}$, proving that combining two fused features with attention mechanism and using self-attention to perform the final fusion do balance two types of depth features.

In Table 6, we explore different depth positional encoding schemes for $f^e_{Dgt}$ and $f^e_{Dbbox}$ within the cross modal attention. From the beginning, we test without any embedding. It is interested to find out that traditional sinusoidal functions for 2D encoding shows a lower precision in 3D object detection, due to lacking abilities representing 3D information.

As shown in the results, the meter-wise encodings represented as $p_D$ approach outperforms other encoding methods by effectively capturing fine-grained depth cues across the range from $d_{min}$ to $d_{max}$, reaching improvements of +3.56% and +3.84% compared to the other two methods. This method enriches the queries with comprehensive scene-level spatial structures, thereby enhancing performance.

To prove the reproducibility of Depth-DETR, we now list out the five runs of MonoDETR and our Depth-DETR in Table 7, it shows that Depth-DETR outperformsMonoDETR in all runs and in the average run.

## CONCLUSION

In this article, we propose a model called Depth-DETR to improve the utilizations of depth encoder in monocular 3D object detection. Depth-DETR employs three encoders focusing on extracting depth features supervised by ground-truth depth and bounding boxes, besides of visual feature. This enables learning better spatial feature information with richer extracted features, and solves the problem of unequal supervision targets in monocular 3D object detection. Depth-DETR also explores fusion methods among different modalities, a cross modal attention mechanism is proposed. With cross modal attention, features with different shapes and dimensions are enable to fuse as one feature, to enrich the expressive abilities of the combined feature. Comprehensive experiments on the KITTI dataset validate that our model achieves outperforms previous competitive monocular-based methods. In the future, we would like to continue working on lightweight Depth-DETR to be able to run in edge devices, as monocular 3D object detection is still the most equipped solutions in vehicles running around the world. In the meanwhile, we will explore our cross modal attentions in fusing LiDAR and images features, which is the most difficult process of multi-modal 3D object detection researches. Also, cross-dataset performance becomes an important ability to concern. Suffered from different camera has different settings, Depth-DETR has not shown excellent performance cross multi datasets, like KITTI, KITTI-360 (*Liao, Xie & Geiger, 2022*) and nuScenes. In the future, we plan to overcome the dependence of aligned LiDAR data of Depth-DETR.

### Funding

This work was supported by the Natural Science Foundation of China (No. 62272093) and the Opening Project of Liaoning Key Laboratory of the Internet of Things Application Technology on Intelligent Construction (No. 6010024037). The funders had no role in

study design, data collection and analysis, decision to publish, or preparation of the manuscript.

## Grant Disclosures

The following grant information was disclosed by the authors:
Natural Science Foundation of China: 62272093.
Opening Project of Liaoning Key Laboratory of the Internet of Things Application Technology on Intelligent Construction: 6010024037.

## Competing Interests

Jie Liu is employed by Anshan Power Supply Company, Liaoning Electric Power Limited Company of State Grid, China. The other authors declare that they have no competing interests.

## Author Contributions

- Zhijian Wang conceived and designed the experiments, performed the experiments, analyzed the data, performed the computation work, prepared figures and/or tables, authored or reviewed drafts of the article, and approved the final draft.
- Jie Liu conceived and designed the experiments, authored or reviewed drafts of the article, and approved the final draft.
- Yixiao Sun performed the experiments, prepared figures and/or tables, and approved the final draft.
- Xiang Zhou performed the experiments, authored or reviewed drafts of the article, and approved the final draft.
- Boyan Sun analyzed the data, authored or reviewed drafts of the article, and approved the final draft.
- Dehong Kong analyzed the data, prepared figures and/or tables, and approved the final draft.
- Jay Xu performed the computation work, prepared figures and/or tables, and approved the final draft.
- Xiaoping Yue performed the computation work, prepared figures and/or tables, and approved the final draft.
- Wenyu Zhang conceived and designed the experiments, authored or reviewed drafts of the article, and approved the final draft.

## Data Availability

The KITTI dataset is available at http://www.cvlibs.net/datasets/kitti.
The code is available at Zenodo: Zhijian, W. (2024). Depth-DETR. Zenodo. https://doi.org/10.5281/zenodo.14259468.

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
