# Peer review of "Applying auxiliary supervised depth-assisted transformer and cross modal attention fusion in monocular 3D object detection"

_PeerJ Computer Science, doi:10.7717/peerj-cs.2656_

## Round 0.1 · original submission · Minor Revisions

Dear authors,

You are advised to critically respond to the reviewer's comments point by point when preparing a new version of the manuscript and while preparing for the rebuttal letter.

Please address all comments/suggestions provided by reviewers, considering that these should be added to the new version of the manuscript.

Kind regards,
PCoelho

Reviewer 1 ·

Basic reporting

The paper proposes Depth-DETR: supervised depth-assisted transformer followed by cross attention fusion for the Monocular 3D detection task. Experiments on the KITTI dataset shows the effectiveness of the approach.

Experimental design

**Generalization and Scaling Up**:
- MonoUni [A] and DEVIANT [B] show the cross-dataset experiments of the KITTI model on the nuScenes frontal val set and report the mean absolute depth error. It would also be good to quantitatively know the cross-dataset performance of MonoUNI, DEVIANT, MonoDETR, and Depth-DETR on this setting.
- A more recent paper SeaBird [C] shows that frontal view detectors do not work well on dataset with large objects such as KITTI-360 [D]. It would be good to benchmark Depth-DETR against SeaBird on KITTI-360 val and leaderboard. (There is already MonoDETR on the KITTI-360 leaderboard)

**Evaluation Protocol and Metrics**:
- The IoU3D threshold of 0.7 is a strict metric and is very sensitive. Please report the KITTI Val results with the threshold of 0.5 of your MonoDETR and Depth-DETR as done in MonoDLE [A] and GUP Net.
- Monocular 3D detection shows significant variation in performance based on the seed than 2D detection, as shown in DEVIANT [B]. Please report the average/median performance across five seeds on the KITTI Val Split of your baseline and Depth-DETR.

**Misc**:
- This paper trains would be difficult to reproduce without the code. Do the authors plan to release the code for all datasets and provide pre trained, especially KITTI test models and logs.

- The paper does not compare with more recent baselines such as DEVIANT [B], MonoRCNN [E], GrooMeD-NMS [F], and MonoGround [G] in Table 1. In other words, several important citations are missing.

References:
- [A] MonoUNI: A Unified Vehicle and Infrastructure-side Monocular 3D Object Detection Network with Sufficient Depth Clues, Jia et al., NeurIPS 2023.
- [B] DEVIANT: Depth Equivariant Network for Monocular 3D Object Detection, Kumar et al., ECCV 2022.
- [C] SeaBird: Segmentation in Bird's View with Dice Loss Improves 3D Detection of Large Objects., Kumar et al., CVPR 2024.
- [D] KITTI-360: A novel dataset and benchmarks for urban scene understanding in 2D and 3D, Liao et al., TPAMI 2022.
- [E] MonoRCNN: Geometry-based Distance Decomposition for Mono3D, Shi et al., ICCV 2021
- [F] GrooMeD NMS: Grouped Mathematically Differentiable NMS for Mono3D, Kumar et al., CVPR 2021
- [G] MonoGround: Homography Loss for Mono3D, Gu et al., CVPR 2022


Validity of the findings

Please see Section 2 for improvements

Additional comments

NA

Reviewer 2 ·

Basic reporting

English needs to be improved. There are several grammatical errors and awkward phrasings (e.g., “existing methods is limited,” “researches have shown,” etc.). A thorough proofreading, especially by someone proficient in academic English, would be beneficial to improve the overall flow.
The introduction provides a good background but could benefit from a more detailed explanation of the knowledge gap. More elaboration on how the proposed model compares to state-of-the-art methods beyond the brief mention in the related works section could strengthen the introduction.

Experimental design

The authors also provide extensive experimental validation using the KITTI dataset, and the results show that the Depth-DETR model outperforms several other leading methods in the field.
Some of the figures could be more clearly labeled, particularly the flow diagrams, to make it easier for readers unfamiliar with the specific technicalities of the field. It would help if the figure captions provided a brief overview of the key insights rather than just describing the components.

Validity of the findings

Overall, the comparative study supported the authors’ claims. The conclusion effectively summarizes the contributions of the paper but could be slightly expanded to include some thoughts on future work, particularly in terms of how the model could be adapted or improved for real-world applications or integrated with other sensor modalities like LiDAR.

Additional comments

The paper presents an approach to improving monocular 3D object detection by proposing the Depth-DETR model, which leverages auxiliary supervised depth-assisted transformers and cross-modal attention fusion. This methodology aims to enhance the prediction of target distances and fuse depth and visual features more effectively, thus addressing some of the limitations in existing monocular 3D object detection systems. The use of two parallel depth encoders supervised by ground truth depth and bounding box data to improve depth estimation is a good contribution. It enhances target distance prediction and provides a competitive advantage over traditional monocular 3D object detection methods. This work outlines the challenges in monocular 3D object detection, specifically the limitations of depth estimation and heterogeneous feature fusion. The authors also provide extensive experimental validation using the KITTI dataset, and the results show that the Depth-DETR model outperforms several other leading methods in the field. demonstrating its competitive edge.

---

## Round 0.2 · accepted · Accept

Dear authors, we are pleased to verify that you meet the reviewer's valuable feedback to improve your research.

Thank you for considering PeerJ Computer Science and submitting your work.

Reviewer 1 ·

Basic reporting

No comment.

Experimental design

No comment.

Validity of the findings

No comment.

Additional comments

I thank the authors for their rebuttal and adding new experiments to the manuscript. The response addresses my concerns. and therefore, I change my rating to accept.

Reviewer 2 ·

Basic reporting

no comment

Experimental design

no comment

Validity of the findings

no comment

Additional comments

The revised work proposes an interesting approach called Depth-DETR to enhance monocular 3D object detection. This method addresses some challenges in depth estimation and multi-modal feature fusion, crucial for improving 3D object detection in autonomous driving. Depth-DETR introduces auxiliary supervised depth encoders that independently process depth data based on ground truth depth and bounding box supervision, mitigating limitations of single supervision methods. By incorporating a cross-modal attention mechanism, the model integrates depth and visual features while maintaining their unique contributions, avoiding the early fusion pitfalls often seen in existing approaches.
The authors proposed a model that outperforms previous methods on the KITTI dataset, achieving an impressive score for moderate difficulty. The improvements highlight its effectiveness in capturing 3D spatial information and addressing depth-related challenges in monocular settings. Despite its strong performance, the study acknowledges areas for further research, such as cross-dataset robustness and reducing reliance on pre-aligned LiDAR data. Overall, Depth-DETR demonstrates good advancements in monocular 3D object detection, presenting the potential for more efficient and precise autonomous driving systems.
The authors have effectively addressed all the concerns raised in the reviews. As a result, the current version is deemed acceptable without the need for further revisions.